# Targeted Metabolome and Transcriptome Analyses Reveal the Pigmentation Mechanism of *Hippophae* (Sea Buckthorn) Fruit

**DOI:** 10.3390/foods11203278

**Published:** 2022-10-20

**Authors:** Jialong Liang, Guoyun Zhang, Yating Song, Caiyun He, Jianguo Zhang

**Affiliations:** 1State Key Laboratory of Tree Genetics and Breeding, Key Laboratory of Tree Breeding and Cultivation, National Forestry and Grassland Administration, Research Institute of Forestry, Chinese Academy of Forestry, Beijing 100091, China; 2Collaborative Innovation Center of Sustainable Forestry in Southern China, Nanjing Forestry University, Nanjing 210037, China

**Keywords:** sea buckthorn, fruit color, chlorophyll, carotenoid, metabolome, transcriptome

## Abstract

The genus *Hippophae* (sea buckthorn) is widely cultivated and consumed in Asia and Europe. The fruit color is an important appearance and commercial trait for sea buckthorn, which is closely related to the biosynthesis and accumulation of various nutrients and pigments. The fruit colors of sea buckthorn are varied, which appear as yellow, orange, red, and brown. However, the nutrients and pigments forming different the fruit colors of sea buckthorn remain unclear. To investigate the mechanism of pigmentation of sea buckthorn fruit, integrative analyses of the transcriptome and targeted metabolome, including the carotenoids, flavonoids, and chlorophylls, were performed in five sea buckthorn varieties with different fruit colors. A total of 209 flavonoids and 41 carotenoids were identified in five sea buckthorn fruits of different colors. The types and contents of flavonoids and carotenoids in the five sea buckthorn fruits were significantly different. Interestingly, we only found a high content of chlorophyll (772.7 mg/kg) in the sea buckthorn fruit with a brown color. The quantities and relative proportions of the flavonoids, carotenoids, and chlorophyll led to the different colors of the sea buckthorn fruits. Using a weighted gene co-expression network analysis (WGCNA), the key genes related to the carotenoids and chlorophyll metabolism were identified. The high content of chlorophylls in the brown fruit was closely related to the downregulated expression of key genes in the chlorophyll degradation pathway, including *SGR*, *SGRL*, *PPH*, *NYC1*, and *HCAR*. Our results provide new insights into the roles of flavonoids, carotenoids, and chlorophylls in the formation of fruit color in sea buckthorn.

## 1. Introduction

*Hippophae* (Sea buckthorn) fruit has been widely consumed as food and traditional folk medicines for centuries, which is rich in nutrients and antioxidants, including protein, vitamins C and E, flavonoids, carotenoids, and organic acids [1,2]. These nutrients and antioxidants in sea buckthorn fruit have been proven to inhibit certain types of cancers, atherosclerosis, cardiovascular diseases, and aging [3,4,5]. The fruit color is an important appearance and quality characteristic for sea buckthorn, and the reagents forming the fruit color are also important nutrients and antioxidants [6,7]. The fruit colors of sea buckthorn are varied, including yellow, orange, red, and brown [8]. Previous studies have revealed that the fruit color was determined by the type and content of pigments, which were mainly chlorophyll, carotenoids, and flavonoids [9,10,11]. During the ripening of the fruit, the fruit color is converted from green (mainly chlorophyll) to various colors in different fruits, such as yellow, orange, red (mainly carotenoids or anthocyanins), blue (mainly anthocyanins), purple, or black (multiple types of pigments) [12,13,14,15,16]. Therefore, an evaluation of the synthesis and accumulation of chlorophyll, carotenoids, and flavonoids would facilitate the comprehension of the mechanism of sea buckthorn fruit pigmentation.

Chlorophylls are essential to photosynthesis, which absorb the blue and red light and reflect green light, meaning they appear green. During fruit ripening, chlorophylls are enzymatically broke down and disappear totally in most of ripe fruit [17]. Other pigments, such as carotenoids and anthocyanins, present various colors in fruit after the degradation of chlorophyll, which may have been present but masked by the chlorophyll before its degradation [10]. However, in some green-ripe fruits, anthocyanins and carotenoids may be present and functional while they are masked by chlorophyll [18,19].

Carotenoids are important natural pigments that exist widely in multiple organs in plants. The conjugated double bonds in carotenoids mainly absorb 400–500 nm wavelength light; therefore, the accumulation of different carotenoids in fruits leads to the yellow, orange, and red fruit colors [20]. In plants, carotenoids also participate in various biological processes, including photoprotection, photomorphogenesis, and plant development [21,22,23]. For humans, carotenoids provide precursors for vitamin A synthesis and serve as antioxidants. The high level of synthesis and accumulation of carotenoids enhance the nutritional and health benefits of fruits [24,25]. The core carotenoid biosynthesis pathway is conserved in the majority of plant species, which starts from isopentenyl diphosphate (IPP) [26,27]. In the carotenoid biosynthesis pathway, a series of pigments with antioxidant activity are produced, including red-colored lycopene; orange-colored carotenes; and yellow-colored lutein, cryptoxanthin, zeaxanthin, antheraxanthin, violaxanthin, and neoxanthin [20].

Flavonoids are widely distributed in plants as important secondary metabolites, which are involved in multiple physiological processes, such as color formation and the protection of plants from UV damage [28]. Flavonoids can be divided into six categories, including flavanols, flavones, flavanones, flavan-3-ols, isoflavones, and anthocyanins, among which anthocyanins commonly serve as deep color pigments for a large number of fruit, which appear as red and turn blue when the pH increases [14,29]. Anthocyanins are synthesized from phenylalanine through the phenylpropanoid and flavonoid pathways, during which a series of intermediate metabolite flavonoids are generated, including flavanones, flavones, dihydroflavonols, and flavonols. While the anthocyanins are unstable, most of the anthocyanins are modified and stored as glycosides.

The aim of this study was to clarify the mechanism of pigmentation in sea buckthorn fruits of different colors. In this study, liquid chromatography tandem mass spectrometry (LC–MS/MS) was performed to detect and quantify carotenoids and flavonoids in five sea buckthorn fruits of different colors, including the orange-colored *Hippophae rhamnoides* subsp. *sinensis* and *H. rhamnoides* subsp. *mongolica* and their hybrid offspring (
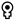
: *H. rhamnoides* subsp. *mongolica* × 
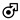
: *H. rhamnoides* subsp. *sinensis*), a red-colored berry “Hongji1” (certificated by National Forestry and Grassland Administration) and yellow-colored berry, as well as the brown-colored *H. neurocarpa* subsp. *neurocarpa*. The chlorophylls were further evaluated using spectrophotometry. The differential regulation of carotenoid, flavonoid, and chlorophyll structural genes was evaluated by analyzing the transcriptional data. Using a weighted gene co-expression network analysis (WGCNA), the co-expression module and key genes involved in carotenoid and chlorophyll metabolism were identified. This study clarifies the role of carotenoids, flavonoids, and chlorophylls in the formation of different colors of sea buckthorn fruits.

## 2. Materials and Methods

### 2.1. Plant Materials and Sampling

Fresh ripe fruits of *H. rhamnoides* subsp. *sinensis* (‘FengNing’, FN) and *H. rhamnoides* subsp. *mongolica* (‘XiangYang’, XY), and their hybrid offspring (
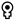
: *H. rhamnoides* subsp. *mongolica* × 
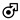
: *H. rhamnoides* subsp. *sinensis*) the red-colored berry “Hongji1” (ZR) and yellow-colored berry (ZY), were harvested from the Experimental Center of Desert Forestry (Dengkou, Inner Mongolia, China) (Figure 1A). These sea buckthorn shrubs were 6 years old and planted on sandy soil with no obvious disease. The fresh ripe fruit of *H. neurocarpa* subsp. *neurocarpa* (Leiguo, LG) was harvested from the Eastern Qinghai–Tibet Plateau (Hongyuan, Sichuan, China). The LG sea buckthorn shrubs were approximately 6–8 years-old and growing in a plateau valley. The fresh ripe sea buckthorn fruits were harvested in August (XY, ZR, and ZY) and September (FN and LG), when the berry growth in terms of size and color was completed and turned soft. Six fruit samples harvested from three individual plants were pooled together as one sample, and three biological replicates were performed for each sea buckthorn variety. The pulp was frozen in liquid nitrogen and stored at −80 °C until analysis.

### 2.2. Flavonoid Extraction and Quantification

The flavonoids were extracted and detected as described previously with some modifications [30]. The frozen sea buckthorn fruits were freeze-dried and then homogenized and powdered using a mixer mill MM 400 (Retsch, Haan, Germany). Here, 100 mg of powder was dissolved in 1 mL 70% aqueous methanol and rotated overnight at 4 °C. The suspension was centrifuged at 10,000 g for 10 min and the supernatant was filtered through CNWBOND Carbon-GCB Solid-Phase Extraction Columns (ANPEL, Shanghai, China) and a 0.22 μm filter (ANPEL, Shanghai, China), and analyzed using HPLC-MS/MS.

The HPLC was performed on a Shim-pack UFLC SHIMADZU CBM30A system (Shimadzu, Kyoto, Japan): column: Waters ACQUITY UPLC HSS T3 C18 column (1.8 µm, 2.1 mm × 100 mm) (Waters, Milford, USA); temperature: 40 °C; injection volume: 5 μL; flow rate: 0.4 mL/min; mobile phase: water with 0.04% acetic acid: acetonitrile with 0.04% acetic acid; gradient programs: 100:0 *v/v* at 0 min, 5:95 *v/v* at 11.0 min, 5:95 *v/v* at 12.0 min, 95:5 *v/v* at 12.1 min, 95:5 *v/v* at 15.0 min. The MS/MS analysis was performed on an API 4500 Q TRAP system (Sciex, Framingham, USA): ion source: turbo spray; temperature: 550 °C; ion spray voltage: 5500 V; curtain gas: 25.0 psi; collision gas: high; declustering potential and collision energy: optimized. The MS/MS data were processed using Analyst 1.6.3 software (Sciex, Framingham, USA). The flavonoids were quantified using the multiple reaction monitoring (MRM) method.

### 2.3. Carotenoid Extraction and Quantification

The carotenoids were extracted and detected as described previously with some modifications [14,31]. The frozen sea buckthorn fruits were freeze-dried and then homogenized and powdered using a mixer mill MM 400 (Retsch, Haan, Germany). Here, 50 mg of powder was dissolved in 1 mL of solution (*n*-hexane/acetone/ethanol (2:1:1, *v/v/v*) with 0.01% butylated hydroxytoluene (BHT)) with internal standard added and vortexed for 20 min at room temperature. The suspension was then centrifuged at 12,000 g for 5 min and the supernatants were collected. The precipitation was then re-extracted as above. All supernatants were dried under a nitrogen gas stream, dissolved in methanol/methyl *tert*-butyl ether (1:1, *v/v*), filtered through a 0.22 μm filter (ANPEL, Shanghai, China), and analyzed by HPLC-MS/MS. 

The HPLC was performed on ExionLC AD system (Sciex, Framingham, USA): column: YMC C30 column (3 µm, 2 mm × 100 mm) (YMC, Devens, USA); temperature: 28 °C; injection volume: 2 μL; flow rate: 0.8 mL/min; mobile phase: (methanol: acetonitrile (3:1, *v/v*) with 0.01% BHT and 0.1% formic acid): (methyl *tert*-butyl ether with 0.01% BHT); gradient programs: 100:0 *v*/*v* at 0 min, 100:0 *v*/*v* at 3 min, 58:42 *v*/*v* at 6 min, 20:80 *v*/*v* at 8 min, 5:95 *v*/*v* at 9.0 min, 100:0 *v*/*v* at 9.1 min, 100:0 *v*/*v* at 11 min. The MS/MS analysis was performed on an API 6500 Triple Quadrupole system (Sciex, Framingham, USA): ion source: APCI+; temperature: 350 °C; curtain gas: 25.0 psi; collision gas: medium; declustering potential and collision energy: optimized. The MS/MS data were processed using Analyst 1.6.3 software (Sciex, Framingham, USA). The carotenoids were quantified using the MRM method, using calibration curves for standards.

### 2.4. Chlorophyll Extraction and Quantification

The chlorophylls were extracted from frozen fruit with 95% ethanol, and the light absorbance values of the extracts were determined at 649 and 665 nm using a spectrophotometer. The chlorophyll concentration in 95% ethanol was calculated as chlorophyll a Ca (mg/L) = 13.95A_665_ − 6.88A_649_, chlorophyll b Cb (mg/L) = 24.96A_649_ − 7.32A_665_, total chlorophyll C = Ca + Cb [32,33]. The chlorophyll content in the fruit (mg/kg) = C × Volume of chlorophyll extract (L)/Weight of fruit (kg).

### 2.5. mRNA Library Preparation and Sequencing

Total RNA samples were extracted from the sea buckthorn fruits using TRIzol reagent (Thermo Fisher, Waltham, USA) and quantified using a NanoDrop Spectrophotometer (Thermo Fisher, Waltham, USA). RNA samples with the RIN number qualified using a Bioanalyzer 2100 (Agilent, Santa Clara, USA) as >7.0 were used for the strand-specific mRNA library construction. In detail, polyadenylated mRNA was captured, fragmented, and reverse-transcribed into cDNA. Then, the dU-labeled double-strand cDNA was synthesized, dA-tailed, and ligated to indexed adapters. The ligated cDNA was digested via heat-labile UDG (NEB, Ipswich, USA) and amplified for 8 cycles to generate the mRNA libraries. The mRNA libraries were sequenced for 150 bp paired-end reads on an Illumina NovaSeq 6000 Sequencer.

### 2.6. Sequence and Primary Analysis

The low-quality bases, undetermined bases, and adaptor contamination were removed using Cutadapt-1.9 software (https://cutadapt.readthedocs.io/en/stable/, accessed on 10 February 2020). The clean reads were aligned to the genome of *H. rhamnoides* subsp. *mongolica* using HISAT2–2.0.4 [34,35]. The aligned reads were assembled using StringTie-1.3.4 with default parameters [36]. Then, all transcriptomes were merged to reconstruct a comprehensive transcriptome using gffcompare-0.9.8 (http://ccb.jhu.edu/software/stringtie/gffcompare.shtml, accessed on 10 February 2020). The gene expression levels were calculated using StringTie and ballgown (http://www.bioconductor.org/packages/release/bioc/html/ballgown.html, accessed on 10 February 2020). The differentially expressed genes (DEGs) were selected with a |Log_2_ (fold change)| > 1 and *p* value < 0.05 using DESeq2 [37]. The DEGs were annotated according to Gene Ontology (GO) terms and Kyoto Encyclopedia of Genes and Genomes (KEGG) pathways [38,39]. The enrichment of specific GO and KEGG pathways among the DEGs was assessed with Fisher’s exact test.

### 2.7. Weighted Gene Co-Expression Network Analysis (WGCNA)

A WGCNA was performed using the R package WGCNA [40]. The edge weight of any two genes was determined by the topology overlap measure provided by the WGCNA. To identify the genes related to pigments accumulation, correlation coefficient of DEGs and differentially accumulated pigments were analyzed. The co-expression network and hub genes were visualized using Cytoscape-3.9.0 [41].

### 2.8. Statistical Analysis

The Pearson correlation coefficient was used to assess the biological replicate correlations of metabolome and transcriptome data. A orthogonal partial least squares discriminant analysis (OPLS-DA) was performed on the identified flavonoids. The differentially accumulated flavonoids were selected with a |Log_2_ (fold change)| ≥ 1, *p* value < 0.05, and VIP (variable importance in project) ≥ 1. For the statistical evaluation of the metabolites, the mean value and standard deviation of the three replicates were calculated, and the significant differences between groups were evaluated via a one-way analysis of variance (ANOVA) at *p* < 0.05 using SPSS software version 20 (SPSS, Chicago, USA). 

## 3. Results

### 3.1. Metabolic Differences among the Fruits of Five Sea Buckthorn Varieties

To investigate the types and contents of pigments involved in the formation of fruit color in sea buckthorn, the carotenoids, flavonoids, and chlorophylls in five varieties of fruit of different colors were measured. The contents and types of flavonoids in sea buckthorn fruits were analyzed using an LC-ESI-Q TRAP-MS/MS system. The correlations between the three biologic replicates were high, with the Pearson correlation coefficient range being between 0.789 and 0.991. A total of 209 kinds of flavonoids were identified in five sea buckthorn varieties, including 56 flavones (48 flavones and 8 flavone C-glycosides), 92 flavonols, 8 flavanols, 5 isoflavones, 10 dihydroflavones, 3 dihydroflavonols, 2 chalcones, 24 tannins, and 9 proanthocyanidins (Appendix A). The numbers of differentially accumulated flavonoids ranged from 45 to 74 between five sea buckthorn varieties (Figure 1B and Appendix A). The anthocyanidins with deep colors, including delphinidin, cyanidin, petunidin, peonidin, malvidin, and pelargonidin, were too small to be detected in all sea buckthorn fruits (Figure 1C). Therefore, the anthocyanidins did not contribute to the differences between the red and yellow fruits in sea buckthorn. The upstream metabolites of anthocyanidins, including leucocyanidin, dihydrokaempferol, and dihydromyricetin, and the downstream metabolites proanthocyanidins were identified in high levels in different sea buckthorn fruits (Figure 1C). A large number of flavonoids, including kaempferol, quercetin, myricetin, and luteolin, present a yellow color, which could provide yellow pigments for the color formation of sea buckthorn fruit (Figure 1C).

To evaluate the contents of carotenoids in sea buckthorn fruits of different colors, we analyzed the five varieties of fruits using LC-APCI-MS/MS. A total of 12 types of carotenoids were identified in the sea buckthorn fruit samples, including α-carotene, β-carotene, γ-carotene, α-cryptoxanthin, β-cryptoxanthin, phytoene, lycopene, lutein, zeaxanthin, antheraxanthin, violaxanthin, and neoxanthin. The ZR fruit contained the highest level of total carotenoids (8096.2 μg/g DW), which was about three-fold the levels in the ZY fruit (2634.7 μg/g DW), XY fruit (2564.6 μg/g DW), and LG fruit (2905.7 μg/g DW), and 14.7-fold that in the FN fruit (550.2 μg/g DW) (Figure 2A). The accumulation of lycopene provided sufficient precursors for the synthesis of downstream carotenoids. The contents of lycopene in the LG fruit and ZR fruit were significantly higher than those in other fruits. During the cyclization of lycopene, the carotenoid synthesis pathway flowed into two branches, the β, ε-branch and β, β-branch. In the β, β-carotene branch, γ-carotene, β-carotene, and β-cryptoxanthin were mainly accumulated in the ZR fruit. The content of zeaxanthin was highest in the ZR fruit, followed by the ZY and XY fruits (Figure 2A). In the β, ε-branch, α-carotene was mainly accumulated in the ZR fruit. The contents of lutein in the ZR and LG fruits were significantly higher than in the others (Figure 2A).

In the ZY fruit, the yellow pigment zeaxanthin was the most abundant carotenoid, which accounted for 88.3% of the total colored carotenoids, followed by the yellow pigment lutein, which accounted for about 10.7% of the total colored carotenoids. The high proportion of yellow pigments and only low amounts of red and orange pigments may result in the yellow fruit color for the ZY fruit. In the ZR fruit, yellow zeaxanthin was still the most abundant carotenoid, which accounted for 58.4% of the total colored carotenoids. Another yellow pigment lutein accounted for about 12.1% of the total colored carotenoids. In addition, the ZR fruit contained high levels of the orange pigment carotenes (11.5% of total colored carotenoids) and the red pigment lycopene (17.0% of total colored carotenoid). In the XY and FN fruits, the proportions of lycopene were between whose in the ZY and ZR fruits, which were 3.5% and 6.7% of the total colored carotenoids, respectively. In brown LG fruit, the proportions of yellow pigment zeaxanthin and lutein were about 9.9% and 37.0%, respectively, while the proportion of lycopene (49.0% of the total colored carotenoids) was higher than in other fruits (Figure 2B). Considering the extremely low contents of anthocyanidins, the distinct accumulation pattern of carotenoids may mainly contribute to the formation of different fruit colors among sea buckthorn varieties.

Chlorophylls are important pigments for the determination of the fruit color in many types of fruits. To evaluate the contents of chlorophylls in sea buckthorn fruits of different colors, we analyzed the fruit samples of five varieties using spectrophotometry. Except for the brown LG fruit, the ripe sea buckthorn fruits only contained small amounts of chlorophylls (Figure 2C), which revealed that most chlorophylls were degraded during fruit ripening. Interestingly, the contents of chlorophylls, including chlorophyll a and chlorophyll b, were extremely high in brown LG fruit (Figure 2C). The total content of chlorophylls in ripe LG fruit (772.7 mg/kg FW) was about 36.1–85.9-fold of that in others (9.0–21.4 mg/kg FW). The high levels of chlorophylls may contribute to the brown color of LG fruit. 

### 3.2. Transcriptome Analysis of the Fruits of Five Sea Buckthorn Varieties

To investigate the molecular basis of the color formation in sea buckthorn fruit, RNA-Seq was performed in the five varieties of ripe fruit. After removing adapter and low-quality sequences, 38,792,620–54,610,674 clean reads for each sample were mapped to the *H. rhamnoides* subsp. *mongolica* reference genome, with read mapping rates in the range of 70.05–93.97% (Appendix A). The Pearson’s correlation analysis showed that the global transcriptome signatures of biological replicates in each group were highly correlated, with the Pearson’s correlation coefficients ranging between 0.909 and 0.979. The overall distribution of the gene expression levels for each sample was analyzed, which was stable between the five varieties (Appendix A). The correlation dendrogram shows that ZR, XY, FN, and ZY are clustered, whereas LG seemed to be an outlier, which indicated that the global gene expression patterns of ZR, XY, FN, and ZY were similar and different from that of LG (Figure 3A).

By analyzing the gene expression levels of all samples, a total of 14,254 DEGs were identified among all groups, and the DEG numbers ranged from 2641 to 8910 between the sample groups (Appendix A). To determine the function of the DEGs, GO and KEGG enrichment analyses were performed (Figure 3B,C). In the cellular component category, the chloroplast, chloroplast stroma, chloroplast envelope, chloroplast thylakoid membrane, chloroplast thylakoid, and thylakoid were the most enriched terms (Figure 3B). In the biological process category, the oxidation–reduction process, thylakoid membrane organization, and photosynthesis were significantly enriched (Figure 3B). In the KEGG analysis, 14,254 DEGs among all groups were annotated to 136 KEGG pathways. In the KEGG pathways, carotenoid biosynthesis, porphyrin and chlorophyll metabolism, and photosynthesis−antenna protein pathways were significantly enriched (Figure 3C), which directly participated in the synthesis and accumulation of carotenoids and chlorophylls. In addition, a series of amino acid metabolism pathways, the glycolysis/gluconeogenesis pathway, and the pentose phosphate pathway were significantly enriched, indicating the differences in amino acids synthesis and carbohydrate metabolism between the five sea buckthorn varieties. 

### 3.3. WGCNA Identified Pigments Accumulation-Related DEGs

To further investigate gene regulation network of pigment synthesis and accumulation in sea buckthorn fruits, a WGCNA was performed using 14,254 DEGs. The DEGs were clustered into 17 co-expressed modules and labeled with different colors (Figure 4A and Appendix A and Appendix A). The expression levels of genes in the same co-expressed module were highly correlated. To identify co-expressed modules related to pigments synthesis and accumulation, the relationships of pigments and co-expressed modules were analyzed (Figure 4B).

The analysis of the module–chlorophyll relationships revealed that the green4, darkorange, and lavenderblush2 modules had high negative correlations with the chlorophylls (correlation coefficient ≤ −0.7 and *p* < 0.05), while the brown2 and indianred3 modules had high positive correlations (correlation coefficient ≥ 0.7 and *p* < 0.05) with the chlorophylls, indicating that the genes in these three modules may play important roles in the high-level accumulation of chlorophylls in LG fruit (Figure 4B). The analysis of the module–carotenoid relationships revealed that the lightsteelblue1 and indianred3 modules had high positive correlations (correlation coefficient ≥ 0.7 and *p* < 0.05) with lycopene, while the tan module had high positive correlations (correlation coefficient ≥ 0.7 and *p* < 0.05) with carotenes (Figure 4B). 

### 3.4. Differential Expression of Chlorophyll Degradation Genes in Five Sea Buckthorn Varieties

To determine the expression pattern of genes in chlorophyll-related modules, the expression levels of these genes were shown in heat maps (Figure 5A and Appendix A and Appendix A). In the green4 module, most genes were expressed at low levels in the LG fruit, exhibiting significant variety specificity (Figure 5A). The GO and KEGG enrichment analyses were performed to investigate the function of the genes in these modules (Figure 5B,C and Appendix A, Appendix A). The GO analysis revealed that the chloroplast, chloroplast stroma, chloroplast envelope, and chloroplast thylakoid membrane were enriched in the green4 module (Figure 5B). In the KEGG analysis, the porphyrin and chlorophyll metabolism pathway was enriched in the green4 module (Figure 5C), which may contribute to the high-level accumulation of chlorophylls. Interestingly, in the green4 module, five hub DEGs were annotated to the chlorophyll degradation pathway, including the chlorophyll synthase (*CLS*, Sph_LG11G002259) gene, chlorophyll b reductase (*NYC1*, Sph_LG0G000805) gene, stay-green/Mg-dechelatase (*SGR*, Sph_LG11G001995) gene, stay-green-like (*SGRL*, Sph_LG4G000009) gene, and stay-green/Mg-dechelatase (*SGR*, Sph_LG2G001906) gene.

To further investigate the expression patterns of chlorophyll-degradation-related genes in sea buckthorn fruit of different colors, the heat maps were performed with the FPKM values of the genes in the chlorophyll degradation pathway (Figure 5D). The stay-green gene (*SGR*), pheophytinase (*PPH*) gene, chlorophyll (ide) b reductase (*NYC1*), chlorophyll synthase (*CLS*), and stay-green-like (*SGRL*) were found to be expressed at low levels in the LG fruit, which was associated with extremely high accumulation levels of chlorophyll a and chlorophyll b in the LG fruit, suggesting that the unique accumulation of chlorophyll a and chlorophyll b in the ripe LG fruit could be the result of impeded chlorophyll degradation.

### 3.5. Expression of Carotenoid and Flavonoid Synthesis Genes in Five Sea Buckthorn Varieties

The type and content of carotenoids played a crucial role in the formation of fruit color of sea buckthorn. The content of carotenes and lycopene were much higher in red color fruit than that in yellow color fruit. To explore the expression pattern of carotenoid-synthesis-related genes in sea buckthorn fruit of different colors, we analyzed the RNA-seq data of ripe fruit of five varieties of sea buckthorn (Figure 6A). Phytoene synthase (PSY) was the first catalytic enzyme for carotenoid synthesis, which catalyzes the synthesis of phytoene and control the synthesis of carotenoids in chromoplasts in fruits [26]. The ζ-carotene desaturase (*ZDS*) gene catalyzed the synthesis of lycopene [26]. *PSY* and *ZDS* were highly expressed in all five sea buckthorn varieties. Lycopene β-cyclase (LCYB) and lycopene ε-cyclase (LCYE) mediate the cyclization of lycopene to synthesize β, β-carotene branch and β, ε-carotene branch carotenoids [20,26]. In the darkorange module, the *LCYE* gene was expressed in a low level in the ZR fruit, which was associated with the high accumulation of lycopene in the ZR ripe fruit. The highly expressed *PSY* and *ZDS*, and low expressed *LCYE* gene may increase the accumulation of lycopene in the ZR fruit. In the LG fruit, the *LCYE* gene was high expressed, which may lead to greater accumulation of the β, ε-carotene branch product lutein and less accumulation of the β, β-carotene branch product zeaxanthin.

In the flavonoid metabolome analysis, anthocyanidins were not detected in the sea buckthorn fruits. To investigate the regulation of flavonoid accumulation in sea buckthorn fruits, the expression levels of genes in the phenylpropane and flavonoid pathways were analyzed (Figure 6B). The colored anthocyanidins were produced from dihydroflavonol, which was catalyzed by dihydroflavonol 4-reductase (DFR) and anthocyanidin synthase (ANS) [26]. Although the anthocyanidins with deep colors were not detected in the sea buckthorn fruits, high expression levels of *DFR* and *ANS* gene were detected. Leucoanthocyanidin reductase (LAR) and anthocyanidin reductase (ANR) catalyzed the synthesis of catechin from leucocyanidin and epicatechin from cyanidin, respectively [15]. The catechin and epicatechin were the initiating monomers of condensed tannins or proanthocyanidin synthesis. The *ANR* gene was expressed at extremely low levels in all sea buckthorn fruit, suggesting that the synthesis of proanthocyanidin in sea buckthorn was mainly catalyzed by LAR.

## 4. Discussion

The fruit color is an important appearance and commercial trait in sea buckthorn. The combined analysis of diverse genetic resources provided crucial information for understanding the molecular basis of the color formation of sea buckthorn fruit. The sea buckthorn plants vary in fruit color, such as yellow, orange, red, and brown. These fruit colors are characteristic of species or specific genotypes. The pigments shaping the fruit colors are important for the fruit quality and flavor and improve the nutritional value of the fruit. The pigments also have health benefits for humans, functioning as antioxidants, having antiaging effects, and inhibiting cardiovascular diseases.

Our results showed that types and contents of carotenoids and chlorophylls mainly contributed to the different fruit colors of sea buckthorn. The anthocyanins with deep colors, however, were too scarce to be detected in the sea buckthorn fruits. The various flavonoids with light yellow colors could provide the yellow pigments to form a basic yellow background color for all five sea buckthorn fruits. In the ripe sea buckthorn fruits with little chlorophyll, the different ratios of lycopene and carotenes were closely associated with the yellow (with a low ratio of lycopene and carotenes), orange, and red (with a high ratio of lycopene and carotenes) fruit colors. The high contents of lycopene and carotenes promoted the nutritional value of the red-colored fruits.

Here, we firstly reported on the stay-green phenotype in ripe sea buckthorn fruit, which was associated with the downregulation of a series of key genes involved in the chlorophyll degradation pathway. The stay-green phenotype was also reported in other fruits in previous studies, such as pepper, oranges, and grapes [42]. Interestingly, in LG fruit, the high accumulation of chlorophylls, combined with the high content of lycopene, formed the brown fruit color.

It is known that in the CMY color system, the different ratios of three colors, including cyan, magenta, and yellow, can produce all other colors [43]. In the CMY color system, the yellow was 0% cyan, 0% magenta, and 100% yellow; the orange was 0% cyan, 50% magenta, and 100% yellow; the red was 0% cyan, 100% magenta, and 100% yellow; the green was 100% cyan, 0% magenta, and 100% yellow; the black was 100% cyan, 100% magenta, and 100% yellow. The different proportions of yellow and red pigments could produce yellow, orange, and red colors. According to the CMY color system, in the yellow fruit ZY, the high ratio of yellow pigments and few red and orange pigments formed the yellow fruit color phenotype. In the orange fruits XY and FN, large amounts of yellow pigment and small amounts of orange and red pigments were mixed to produce the orange fruit colors. In the red fruit ZR, a large amount of yellow pigment and large amounts of orange and red pigments were mixed to produce the red fruit color (Figure 7). It is worth noting that the mixing of different ratios of red and green pigments produce different colors, including brown, deep green, reddish brown, and black. In the brown fruit LG, large amounts of orange and red pigments and a large amount of green pigment were mixed to produce the brown fruit color (Figure 7).

In the LG fruit, the high accumulation of chlorophylls was negatively associated with low expression levels of genes, including the stay-green gene (*SGR*), pheophytinase (*PPH*) gene, chlorophyll(ide) b reductase (*NYC1*), chlorophyll synthase (*CLS*), and stay-green-like (*SGRL*), which participated in the chlorophyll degradation progress. The chlorophyll degradation pathway has been studied and important functional genes for chlorophyll degradation have been characterized [17]. The conversion of chlorophyll b to chlorophyll a is the first step for chlorophyll b degradation. The chlorophyll (ide) b reductase (*NYC1*) gene played a crucial role in the conversion of chlorophyll b to chlorophyll a [44]. The suppression of *NYC* in Arabidopsis resulted in a cosmetic stay-green phenotype [44]. In the chlorophyll a degradation pathway, stay-green/Mg-dechelatase (*SGR*) removes the magnesium ion from chlorophyll a to produce pheophytin a [17]. The pheophytinase (*PPH*) gene encoded a genuine in vivo phytol hydrolase, which removed phytol from pheophytin, mediating the conversion of pheophytin a to pheophorbide a [17]. Therefore, the downregulated expressions of key genes involved in the chlorophyll degradation pathway could lead to the inhibited conversion of chlorophyll b to chlorophyll a and chlorophyll degradation, which are consistent with the extremely higher accumulation of chlorophyll a and chlorophyll b in ripe LG fruit (Figure 7).

The gene expression pattern in the carotenoid synthesis pathway participated in the formation of an accumulation pattern for the carotenoids. The high expression of *PSY* and *ZDS* in the sea buckthorn fruit provided sufficient precursors for the downstream carotenoid synthesis. The *LCYE* and *LCYB* genes regulate the synthesis of downstream products from lycopene. The downregulation of *LCYE* in the ZR fruit limits the synthesis of downstream carotenoids from lycopene. The highly expressed *PSY* and *ZDS* genes and poorly expressed *LCYE* gene may promote the accumulation of lycopene in the ZR fruit, providing the main red pigment for sea buckthorn fruits (Figure 6A). Capsanthin–capsorubin synthase (CCS) catalyzed the conversion of antheraxanthin and violaxanthin to capsanthin and capsorubin, respectively [14]. All sea buckthorn fruits expressed extremely low levels of the *CCS* gene, which was consistent with the low contents of capsanthin and capsorubin in the sea buckthorn fruits. The characterization of these key genes in the carotenoid synthesis pathway provided the foundation for the breeding of sea buckthorn varieties with different fruit colors.

Although the anthocyanidins with deep color were too scarce to be detected in the sea buckthorn fruits, the accumulation pattern of the flavonoids and the gene expression profile in the phenylpropane and flavonoid pathways were characterized. Dihydroflavonols, including dihydrokaempferol, dihydroquercetin, and dihydromyricetin, were upstream precursors for anthocyanidin synthesis. Dihydroflavonols and the downstream metabolite procyanidins were detected in different contents in ripe sea buckthorn fruits, suggesting that anthocyanidins may be synthesized and soon converted to downstream metabolites in sea buckthorn fruits. The artificially cultivated anthocyanidin-enriched fruits, through genetic engineering methods or genotype selection arising from natural breeding, have been widely planted [45]. The study of flavonoid metabolism in sea buckthorn fruits would facilitate the cultivation of anthocyanidin-enriched sea buckthorn.

## 5. Conclusions

In conclusion, targeted metabolome and transcriptome comparisons were performed in sea buckthorn varieties with different fruit colors. The yellow-colored flavonoids, lutein, and zeaxanthin could provide the yellow pigment for the sea buckthorn fruits. The yellow fruit ZY was mainly colored by zeaxanthin, lutein, and flavonoids, while the contents of carotenes and lycopene were extremely low. In the red fruit ZR, in addition to the yellow-colored zeaxanthin, lutein, and flavonoids, high levels of lycopene and carotenes were identified, which may result in the red color phenotype of the fruit. In the brown LG fruit, the low expression level of chlorophyll degradation genes led to the significantly high accumulation level of chlorophylls. Combined with the high content of the red pigment lycopene, the LG fruit displayed a brown color. Together, this study provides novel insights into the roles of the synthesis and accumulation of flavonoids and carotenoids and the degradation of chlorophyll in the pigmentation of sea buckthorn fruit, but also lays a solid biological foundation for the breeding of high-quality nutritious sea buckthorn.

## Figures and Tables

**Figure 1 foods-11-03278-f001:**
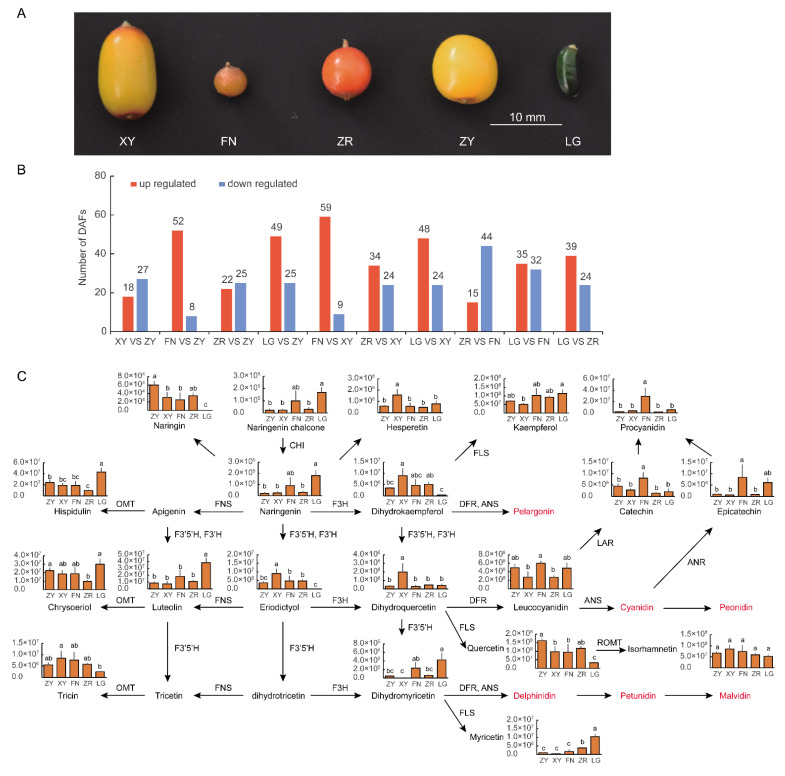
Fruit colors and quantification of flavonoids in the fruits of five sea buckthorn varieties: (**A**) colors of ripe sea buckthorn fruit: XY, FN, ZR, ZY, LG; (**B**) numbers of differentially synthesized flavonoids between five sea buckthorn varieties; (**C**) flavonoid levels in the fruits of five sea buckthorn varieties. Each bar represents the average of three biological replicates plus the standard deviation. The significant difference between the data were evaluated via one-way ANOVA and labeled using different letters (*p* < 0.05).

**Figure 2 foods-11-03278-f002:**
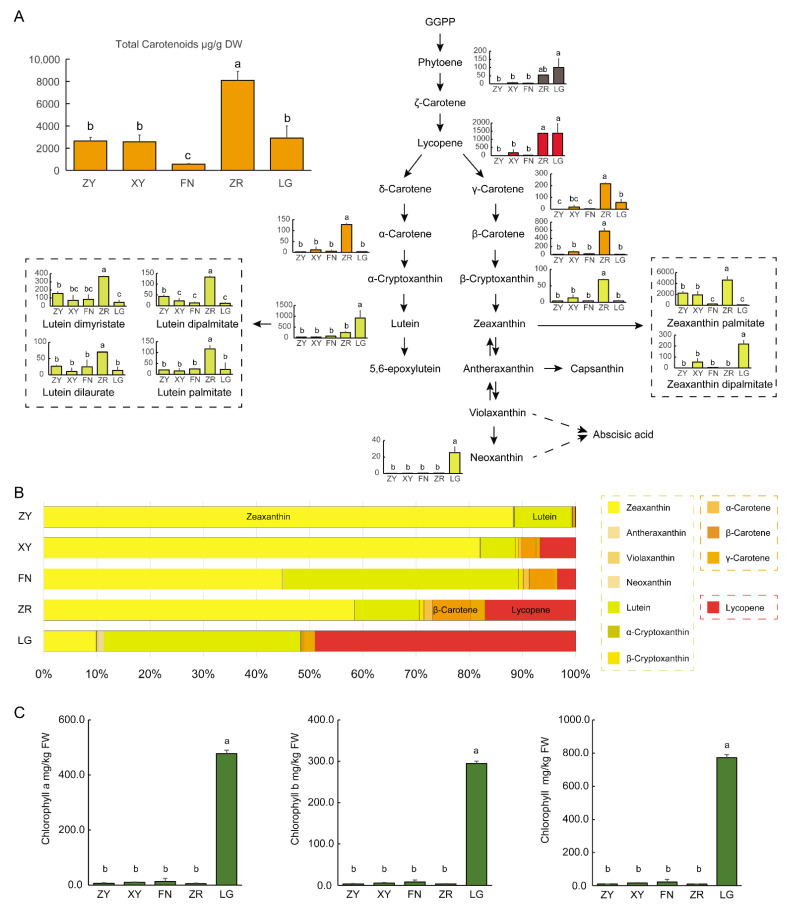
Quantification of carotenoids and chlorophylls in the fruits of five sea buckthorn varieties: (**A**) the carotenoid levels in the fruits of five sea buckthorn varieties; (**B**) the proportions of each colored carotenoid of the total colored carotenoids in the fruits of five sea buckthorn varieties; (**C**) the levels of chlorophylls a and b and the total chlorophylls in the fruits of five sea buckthorn varieties. Each bar represents the average of three biological replicates plus the standard deviation. Significant difference between the data were evaluated via a one-way ANOVA and labeled with different letters (*p* < 0.05).

**Figure 3 foods-11-03278-f003:**
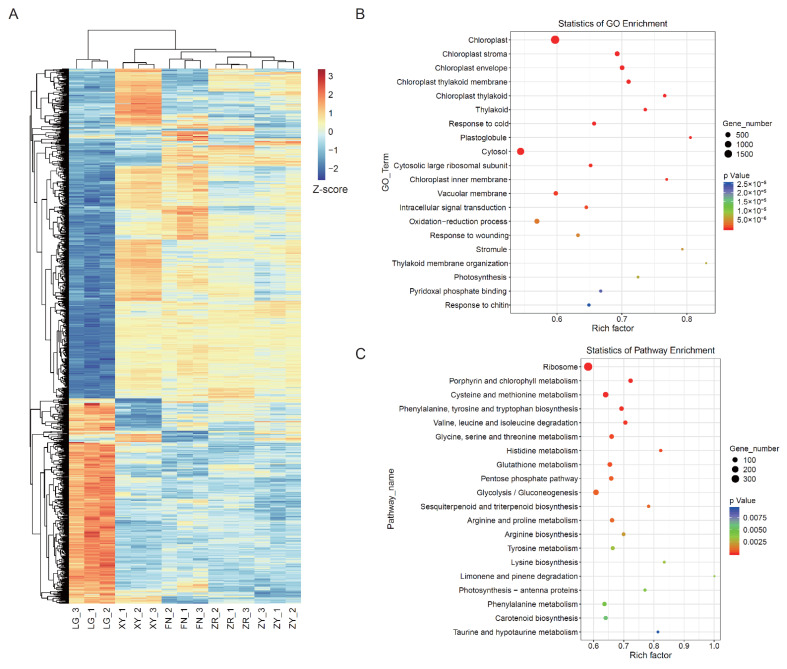
Global characterization of gene expression patterns of ripen sea buckthorn fruit of different colors: (**A**) cluster dendrogram showing the relationship of global gene expression patterns of five sea buckthorn fruits; (**B**) the GO enrichment analysis of the differently expressed genes in the fruits of five sea buckthorn varieties; (**C**) the KEGG enrichment analysis of the differently expressed genes in the fruits of five sea buckthorn varieties.

**Figure 4 foods-11-03278-f004:**
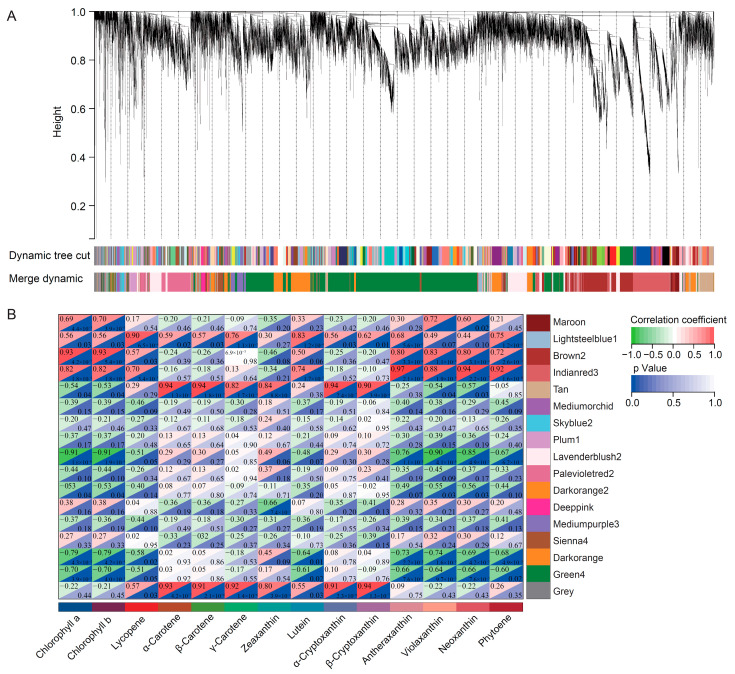
A weighted gene co-expression network analysis of DEGs in ripe fruits of five sea buckthorn varieties: (**A**) cluster dendrogram showing the gene co-expression modules from the WGCNA analysis; (**B**) module–chlorophyll and module–carotenoid relationships. Each row represents a module. Each column represents a specific chlorophyll or carotenoid. The value in each cell represents the correlation coefficient between the module and the chlorophyll or carotenoid, as well as the *p* value.

**Figure 5 foods-11-03278-f005:**
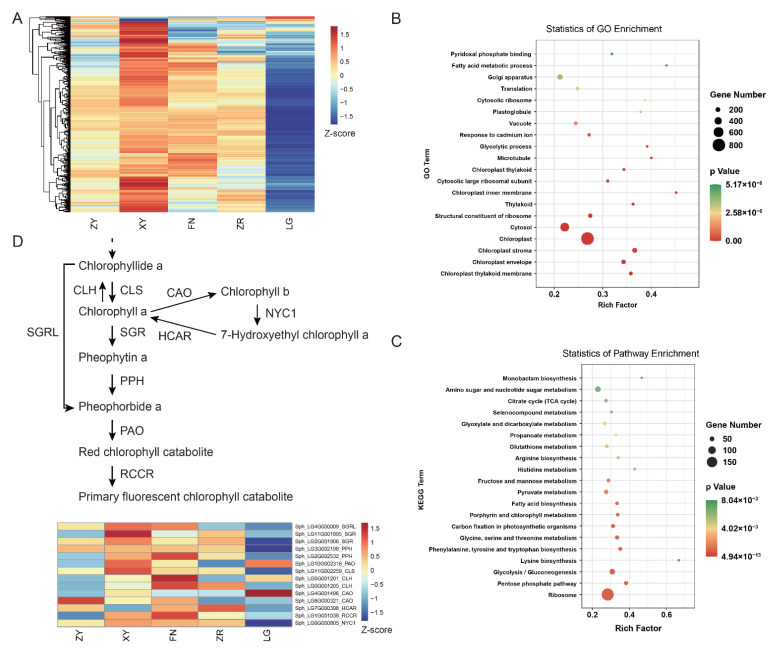
Chlorophyll-accumulation-related genes analysis: (**A**) heat map of genes in the green4 module; (**B**) the GO enrichment analysis of genes in the green4 module; (**C**) the KEGG enrichment analysis of genes in the green4 module; (**D**) chlorophyll degradation pathways and genes. *CLS*, chlorophyll synthase; *CLH,* chlorophyllase; *SGR*, stay-green/Mg-dechelatase; *SGRL*, stay-green-like; *PPH*, pheophytinase; *PAO,* pheophorbide a oxygenase; *RCCR,* red chlorophyll catabolite reductase; *CAO*, chlorophyllide a oxygenase; *NYC1,* chlorophyll b reductase; *HCAR,* 7-hydroxymethyl chlorophyll a reductase. The pathway was constructed based on the KEGG pathway and references. Heat map of gene expression levels based on FPKM values.

**Figure 6 foods-11-03278-f006:**
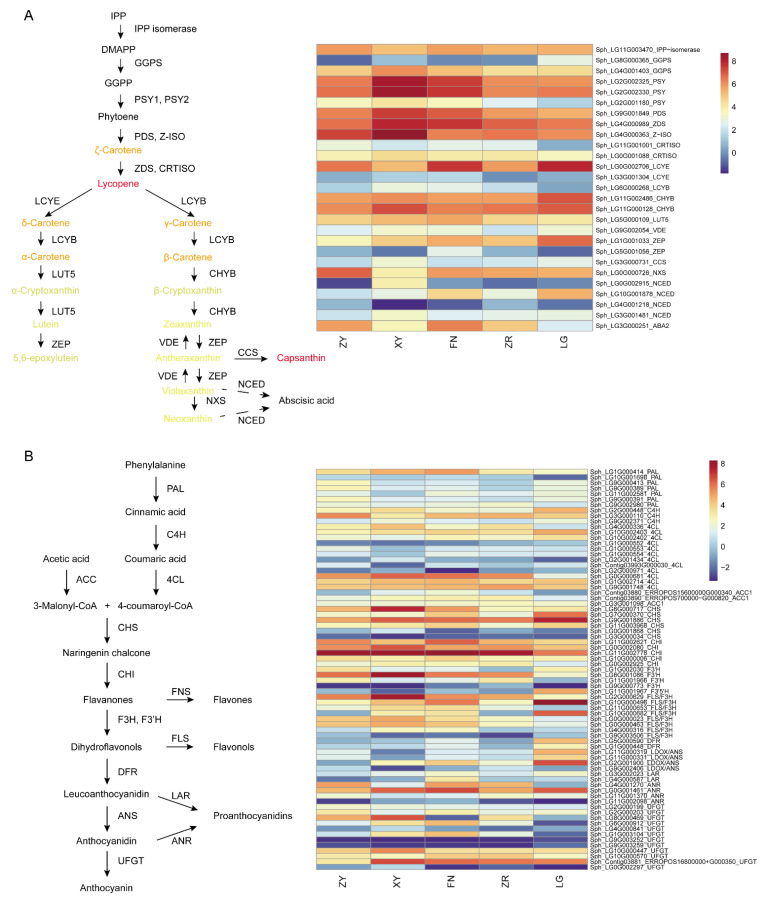
Carotenoid and flavonoid metabolism pathways and synthesis genes. (**A**) Carotenoid metabolism pathways and synthesis genes. IPP, isopentenyl diphosphate; DMAPP, dimethylallyl pyrophosphate; GGPP, geranylgeranyl diphosphate; *GGPS*, GGPP synthase; *PSY*, phytoene synthase; *PDS*, phytoene desaturase; *Z-ISO*, ζ-carotene isomerase; *ZDS*, ζ-carotene desaturase; *CRTISO*, carotenoid isomerase; *LCYE*, lycopene ε-cyclase; *LCYB*, lycopene β-cyclase; *CHYB*, β-carotene hydroxylase; LUT5, Lutein deficient 5; *ZEP*, zeaxanthin epoxidase; *VDE*, violaxanthin de-epoxidase; *CCS*, capsanthin–capsorubin synthase; *NXS*, neoxanthin synthase; *NCED*, 9-cis-epoxycarotenoid dioxygenase. (**B**) Flavonoid metabolism pathways and synthesis genes. *PAL*, phenylalanine ammonia lyase; *C4H*, cinnamate 4-hydroxylase; *4CL*, 4-coumarate-CoA ligase; *ACC1*, acetyl-CoA carboxylase 1; *CHS*, chalcone synthase; *CHI*, chalcone isomerase; *F3H*, flavanone 3-hydroxylase; *F3′H*, flavonoid 3′-hydroxylase; *DFR*, dihydroflavonol 4-reductase; *ANS*, anthocyanidin synthase; *UFGT*, flavonoid-3-O-glucosyltransferase; *FNS*, flavone synthase; *FLS*, flavonol synthase; *LAR*, leucoanthocyanidin reductase; *ANR*, anthocyanidin reductase. The pathway is constructed based on KEGG pathway and references. Heat maps of gene expression levels based on FPKM values.

**Figure 7 foods-11-03278-f007:**
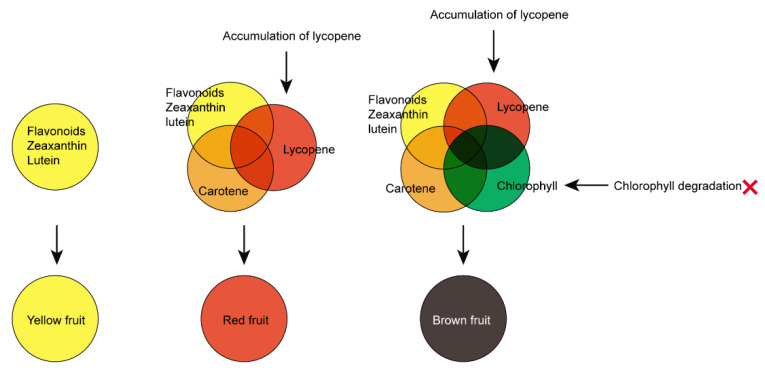
The regulation pattern of pigments and the CMY model of sea buckthorn fruit color formation.

## Data Availability

The data reported in this study are available from the corresponding author upon reasonable request.

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
