# Peer review of "Targeted Metabolome and Transcriptome Analyses Reveal the Pigmentation Mechanism of Hippophae (Sea Buckthorn) Fruit"

_foods, 2022, doi:10.3390/foods11203278_

Round 1

Reviewer 1 Report

Ms. Ref. No.: foods-1968991

Title:  Targeted metabolome and transcriptome analyses reveal the pigmentation mechanism of Hippophae L. fruit

Dear editor

This study aimed to clarified the role of carotenoids, flavonoids, and chlorophylls in formation of colors of Hippophae L. fruits.

The writing is good, but some verb need to change (from past tense to present tense for example lines 14, 15, 17, 35, 38, 40, 41, 43, 49, 51, 55, 57, 64, 67, 70, 74, 76, 79, 247).

The experiment has been set up well, but the data analysis section is not provided.

Besides some errors exist in manuscript as mentioned below.

Comments to Author

Title:

The full scientific name of plant should provided

Abstract:

The result section of abstract is qualitative, please add some important data.

Key words

should capitalize each word.

Introduction

For first refer to plant name please state full scientific name, but for other case just the English name (without the name of botanist)

Material and methods

Add references for 2.2, 2.3 and 2.4 sections

Why the unit for chlorophyll in material and methods is not same as result section?

Data analysis section is missed

Results

Figure 1.B is better to delete

Figure 3 A is better to delete

Discussion

The titles of Lines 395, 418, 439, 457 better to delete

It is suggested to move figure 7 to result section and not refer to any figure or table.

Reviewer 2 Report

I have only a few, in the first place, technical corrections, see the enclosed file, please.

Reviewer 3 Report

English editing is highly recommended. 

Ripe stage is very subjective and the stages can affect the findings of this study significantly.   Please include objective / quantitative to indicate the 'ripe stage' of fruit.  

Please include the exact age of plants and harvesting time of fruit as these  can affect fruit metabolites.  

The agronomic practices and geographical condition of the plants should be included as these able to affect metabolites of fruits. 
